

# Covariance-based selection of parameters for particle filter data assimilation in soil hydrology

Alaa Jamal[1] and Raphael Linker [2]

[1]Department of Civil and Environmental Engineering, University of Illinois at Urbana-Champaign; 205 N Mathews Ave, Urbana, Illinois.

[2] Faculty of Civil and Environmental Engineering, Technion- Israel Institute of Technology; Derech Ya'akov Dori, Haifa, Israel.

*Correspondence to*: Alaa Jamal (ajamal@illinois.edu).

**Abstract.** Real-time in-situ measurements are increasingly used to improve the estimations of simulation models via data assimilation techniques such as particle filter. However, models that describe complex processes such as water flow contain a large number of parameters while the data available is typically very limited. In such situations, applying particle filter to a large, fixed set of parameters chosen a priori can lead to unstable behavior, i.e. inconsistent adjustment of some of the parameters that have only limited impact on the states that are being measured. To prevent this, in this study correlation-based variable selection is embedded in the particle filter, so that at each data assimilation step only a subset of the parameters is adjusted. More specifically, whenever measurements become available, the most influential (i.e., the most highly correlated) parameters are determined by correlation analysis, and only these are updated by particle filter. The proposed method was applied to a water flow model (Hydrus-1D) in which states (soil water contents) and parameters (soil hydraulic parameters) were updated via data assimilation. Two simulation case studies were conducted in order to demonstrate the performance of the proposed method. Overall, the proposed method yielded parameters and states estimates that were more accurate and more consistent than those obtained when adjusting all the parameters.

## 1 Introduction

Accurate and proper estimation of prognostic variables (e.g. soil moisture) has been receiving increasing attention in the past years. The mathematical models that describe such complex processes (e.g. Hydrus (Simunek et al., 1998)) contain a large number of parameters. Calibrating such models, which are non-linear, is far from trivial, especially since in real settings the data available for the task is limited. Therefore, real-time in-situ measurements are increasingly used to improve the estimations of such simulation models via data assimilation techniques (Das & Mohanty, 2006; Das et al., 2008; Brandhorst et al., 2017; Abbaszadeh et al., 2018; Bauser et al., 2018; Berg et al., 2019; Jamal and Linker, 2019).

One of the most widely used data assimilation (DA) methods is the ensemble Kalman filter (EnKF) (Reichle et al., 2002; De Lannoy et al., 2007; Jamal and Linker, 2019). Despite the popularity of EnKF, this technique requires determining some factors, such as covariance inflation and covariance localization, which influence strongly the behavior of the EnKF. To



overcome these limitations, particle filer (PF) has been increasingly used as an alternative data assimilation method (DeChant and Moradkhani, 2012; Yin and Zhu, 2015; Abbaszadeh et al., 2018; Berg et al., 2018; Jamal and Linker, 2020).

A main drawback of PF is particle degeneracy as some particles become associated with negligible weights. Several methods have been suggested to mitigate this problem. Moradkhani et al. (2005), who worked with a hydrological model, suggested perturbing the parameters using Gaussian noise. Another method suggested to handle weight degeneracy is the combination

of PF with Monte Carlo Markov Chain (MCMC) (Andrieu et al., 2010; Moradkhani et al., 2012). This integration helps the PF to replace low probability particles with particles that have higher probabilities. On the other hand, intelligent search and optimization methods, such as Genetic algorithm (GA), have been used as well to alleviate the degeneracy problem. Jamal and Linker (2020) showed the usefulness combining MCMC and PF with Genetic operators of GA (named GPFM). This approach was applied to simultaneous estimation of state variables and parameters in a crop growth model.

As mentioned above, the models that describe complex processes such as water flow and/or crop development (e.g. SWAP (Kroes et al., 2017)) contain a large number of parameters and the data available is rarely sufficient for calibrating all the parameters at once. In such cases, an sub-set of the most influential parameters should be calibrated rather than attempting to calibrate all the parameters at once. This process of parameters selection can be performed beforehand via sensitivity analysis (Hamby, 1994; Della Peruta et al., 2014; Xu et al., 2016; Wu et al., 2019). However, the results of the sensitivity

analysis can be somehow misleading or irrelevant since the "importance" of a specific parameter depends on the state of the system. For instance, in soil hydrology, the residual water content has a strong influence on the model behavior during dry periods while its value is basically irrelevant during wet periods. Similarly, hydraulic conductivity can be more influential during wet periods than during dry ones (Pue et al., 2019). Therefore, the calibration of the residual water content during wet periods is at best unnecessary and can even cause a degradation of the estimated value (Jamal and Linker, 2020). Therefore,

when data assimilation (rather than off-line batch calibration) is considered, the sensitivity of the parameters should be determined dynamically in real-time. While performing "true" sensitivity analysis in real-time is not possible due to its high computational burden, an alternative is to use correlation analysis (Manache and Melching, 2008) for quantifying the sensitivity of the parameters. The concept of correlation analysis is to evaluate the correlation between the updated variables and the measured ones, which in turn can be used to identify the most influential parameters.

The usage of correlation analysis for parameter selection in data assimilation is limited. However, correlation analysis is used somehow in Kalman filters through the Kalman gain calculation. Hu et al. (2019), who reported data assimilation in a crop model, suggested to use states correlations, in addition to the traditional use of correlation analysis for determining the Kalman gain, in order to prevent updating poorly correlated states. State variables with low correlation to the measurements (or measured states) are updated less frequently than the highly correlated ones.

In traditional particle filters, the whole set of parameters is updated, regardless of the sensitivity or correlation of each parameter inside this set to the available measurements (Moradkhani et al., 2012; Berg et al., 2019). This study presents a novel particle filter in which correlation-based variable selection is embedded. Whenever measurements become available, the most influential (i.e., the most highly correlated) parameters are determined by correlation analysis, and only these





parameters are updated by PF. More specifically, the data assimilation technique of genetic - operators based PF with Monte

Carlo Markov Chain (based on Jamal and Linker (2020)) is combined with calculation of the correlation between each parameter and each measured state. Then, only highly correlated parameters are involved in the selection, mutation, crossover and resampling operations of the PF, leading to the novel approach denoted C-GPFM. The proposed method is applied to a water flow model (Hydrus-1D) in which states (soil water contents) and parameters (soil hydraulic parameters) are updated via data assimilation. Two case studies were generated and analyzed in order to investigate the performance of

the proposed method. The results of the proposed method are compared to the case where the whole set of parameters is updated.

## 2 Methods

### 2.1 Particle filtering

Nonlinear dynamic systems are often described by the following finite difference equations:

$$x_t = f(x_{t-1}, u_{t-1}, \theta_{t-1}) + \omega_t \tag{1}$$

$$y_t = h(x_t) + v_t \tag{2}$$

where $f(\ )$ denotes the model, $h(\ )$ denotes the measurements operator, $x_t \in \mathbb{R}^n$ denotes the state vector at time $t$, $u_t$ is the (uncertain) forcing input, $\theta \in \mathbb{R}^d$ is the vector of model parameters, $y_t \in \mathbb{R}^m$ is the vector of measured variables. $\omega_t$ and $v_t$ are the process and measurements noise, respectively, which are assumed to be white noises with zero mean and covariance $Q_t$ and $R_t$, respectively. In addition, they are assumed to be independent. Based on Bayesian estimation, given a measurement $y_t$ at time $t$ the posterior distribution of the state at time $t$ is as follows:

$$p(x_t|y_t) = p(x_t|y_{1:t-1}, y_t) = \frac{p(y_t|x_t)p(x_t|y_{1:t-1})}{p(y_t|y_{1:t-1})} = \frac{p(y_t|x_t)p(x_t|y_{1:t-1})}{\int p(y_t|x_t)p(x_t|y_{1:t-1})dx_t} \tag{3}$$

$$p(x_t|y_{1:t-1}) = \int p(x_t, x_{t-1}|y_{1:t-1})dx_t = \int p(x_t|x_{t-1})p(x_{t-1}|y_{1:t-1})dx_{t-1} \tag{4}$$

where $p(y_t|x_t)$ is the likelihood of the observed measurement given the estimated state at time $t$, $p(x_t|y_{1:t-1})$ is the prior distribution of the state, and $p(y_t|y_{1:t-1})$ is a normalization factor. The marginal likelihood function $p(y_{1:t})$ can be computed as:

$$p(y_{1:t}) = p(y_1)\prod p(y_t|y_{1:t-1}) \tag{5}$$

The analytical solution of (3) cannot be obtained due to the non-linearity of the process and the multi-dimensionality of the problem. In procedures based on particle filter, the posterior distribution is approximated using an ensemble of particles, that

is by running an ensemble of models in parallel. The reader is referred to Moradkhani et al., (2005) for a more detailed description of particle filters.

The specific PF procedure implemented in this work was the Genetic Operator-Based Particle Filter Combined with Markov Chain Monte Carlo (GPFM), described in Jamal and Linker (2020). This procedure integrates GA operators and MCMC





within the PF. Notice that the final resampling operation, which was defined as optional in Jamal and Linker (2020), was
implemented in the present study via stochastic universal resampling (Kitagawa, 1996).

## 2.2 Correlation Analysis

As mentioned in the Introduction, the main novelty in this work is the combination of correlation analysis with the particle
filter, in-real time. The purpose is to involve only a subset of the parameters in the crossover and mutation operations, in the
MCMC and in the resampling process steps of the GPFM procedure.


In order to determine the parameters most highly correlated to the available measurements, the correlation coefficient
between each parameter and each measured state is calculated as follows:

$$cor_t^{i,j}, = \left| \frac{1}{N-1} \frac{\sum_{k=1}^{N} (\theta_{i,t}^k - \bar{\theta}_{i,t})(h(x_t^k)_j - \overline{h(\bar{x}_t)_j})}{\sigma_{\theta_{i,t}} \sigma_{h(x_t)_j}} \right| \tag{6}$$

$$\bar{\theta}_{i,t} = \frac{\sum_{k=1}^{N} \theta_{i,t}^k}{N}, \quad \overline{h(\bar{x}_t)_j} = \frac{\sum_{k=1}^{N} h(x_t^k)_j}{N}$$

where $\theta_{i,t}^k$ denotes the $i$-th parameter in particle $k$ at time $t$, $h(x_t^k)_j$ denotes the $j$-th predicted measured state in particle $k$ at
time $t$ and $N$ is the ensemble size. $\sigma_{\theta_{i,t}}$ and $\sigma_{h(x_t)_j}$ are the standard deviations of the $i$-th parameter and $j$-th predicted output
at time $t$ which are calculated as follows:

$$(7)$$

$$\sigma_{\theta_{i,t}} = \sqrt{\frac{\sum_{k=1}^{N} (\theta_{i,t}^k - \bar{\theta}_{i,t})^2}{N-1}},$$

$$\sigma_{h(x_t)_j} = \sqrt{\frac{\sum_{k=1}^{N} (h(x_t^k)_j - \overline{h(\bar{x}_t)_j})^2}{N-1}}$$

The correlation coefficient ranges from 0 to 1, corresponding to weak and strong correlation, respectively. After the
calculation of the correlation coefficients, the weighted average of the correlation coefficients for each parameter is
calculated in order to take into account the correlation of the parameter with all of the measurements. These weights are
dictated by the importance of the measurements (or measured states) in the particles weights, i.e. the larger the value of the
weight of the measurement in the particle weights, the larger its weight in the correlation coefficient. Therefore, the weight
of each measured state correlation coefficient was chosen according to the importance of the measurement in the particles
weights calculation, as follows:

$$(8)$$

$$cor_t^i = \sum_{j=1}^{M} W_t^j cor_t^{i,j}$$





$$W_t^j = \frac{\sum_{i=1}^{N} L\left(y_t^j \middle| x_t^i, \theta_t^i\right)}{\sum_{j=1}^{M} \sum_{i=1}^{N} L\left(y_t^j \middle| x_t^i, \theta_t^i\right)}$$

where $cor_t^i$ is the correlation coefficient of parameter $i$ at time $t$ and $L$ denotes the likelihood function. Noteworthy, $cov_i$ is

still between 0 and 1. It is worth mentioning that the correlation coefficients are calculated based on the prior ensemble (i.e. the particles before performing the crossover and mutation operations) since the proposal parameters are directly influenced by crossover and mutation while the prior ensemble is more physically driven.

After the calculation of $cor_i$ for each parameter, the parameters are ranked from highly influential to low importance based on their correlation coefficients and only the parameters with correlation coefficients above some threshold are chosen for

crossover, mutation, MCMC and resampling. Figure 1 describes the overall framework. The first step (Step A) is to perform one-step ahead predictions with the whole ensemble and compute the weight associated with each particle. The second step is to calculate the correlation coefficients and select the most influential parameters (step B). The third step (Step C) is to select particles by roulette wheel selection. Crossover and mutation are then applied on the most influential parameters from Step B (Step D). One-step predictions are performed with the new ensemble (Step E) and the results of each pair of

prior/proposal particles are compared to determine which particle must be retained (Step F). Finally, resampling is applied (Step G) over the state and most influential parameters to generate the new particles.



Initial particles
at t

**C**
Choose parents chromosomes
by roulette wheel selection

**A**
Run ensemble
to t + 1

Get prior at t + 1

Calculate weight for
each particle

**D**
Crossover and mutation
for the state and influential
parameters in the selected
particles

**B**
Calculate correlation
coefficients and select the
influential parameters

**E**
Run ensemble
to t + 1

Get proposal prior
at t + 1

Calculate weights for
each particle

**F**
Calculate the metropolis
acceptance ratio for
each particle

metropolis
acceptance ratio
> threshold

Yes

No

Keep proposal state
and influential
parameters particles

Keep prior state and
influential parameters
particles

**G**
Resampling for the state
and most influential paramters

Get posterior for
t + 1





**Figure 1: the overall framework.**

## 3 Case studies

The method described above was tested with a synthetic case study of 1-D soil profile with three layers. Two models were created, denoted 'true' and 'biased'. Simulations with the "true" model were performed to generate synthetic measurements and the data assimilation procedure was applied to a model which was initialized as the "biased" model. The estimations of the parameters was restricted to an interval of $\pm 40\%$ around the values of the parameters of the "biased" model. The soil hydraulic properties of the two models are described in Table 1. The parameter values were such that the flow is highly

dynamic and therefore the problem is challenging. The lower bound condition was set to free drainage, and the initial conditions were set as 0.35 and 0.20 soil water content for the whole profile for the true and biased models, respectively. Soil water content at 18 cm and 42 cm were "measured" daily with white gaussian noise with standard deviation of 1%.

**Table 1:** *The soil parameters used in the case studies.*

| Parameter | Description | Depth | True | Biased |
|---|---|---|---|---|
| $\theta_{sat}$ | Saturated water content | 0-20 cm | 0.43 | 0.48 |
| | | 21-40 cm | 0.41 | 0.36 |
| | | 41-60 cm | 0.43 | 0.48 |
| $\alpha$ | Air entrance value parameters | 0-20 cm | 2.68 | 2.1 |
| | | 21-40 cm | 2.10 | 2.5 |
| | | 41-60 cm | 2.68 | 2.1 |
| $K_{sat}$ | Saturated hydraulic conductivity [$cm/day$] | 0-20 cm | 713 | 613 |
| | | 21-40 cm | 230 | 270 |
| | | 41-60 cm | 713 | 613 |
| $\theta_{res}$ | Residual water content | 0-20 cm | 0.045 | |
| | | 21-40 cm | 0.061 | |
| | | 41-60 cm | 0.045 | |
| $n$ | Shape parameter | 0-20 cm | 0.14 | |
| | | 21-40 cm | 0.10 | |
| | | 41-60 cm | 0.14 | |

The ensemble size $N$ was chosen as 100. The crossover factor for state and parameters were set to 0.01 and 0.05,

respectively. For the mutation for the state and parameters, 10% standard deviation was chosen. Larger crossover was set for the parameters than for the state in order to favor improvement of the model's dynamics over mere instantaneous adjustment of the state. The threshold of correlation in Equation 8 was set arbitrarily to 0.2, which corresponds to a non-negligible linear relationship (Akoglu, 2018).





A first study was conducted with random precipitation as boundary condition, which mimics a typical rainy period. A second
analysis was carried out with boundary conditions consisting of ten days irrigation-drainage cycles. This analysis was
conducted to evaluate the performance of the method under clearly defined variations of soil moisture (and therefore
influential parameters). The boundary condition of the two case studies is described in Fig. 2.

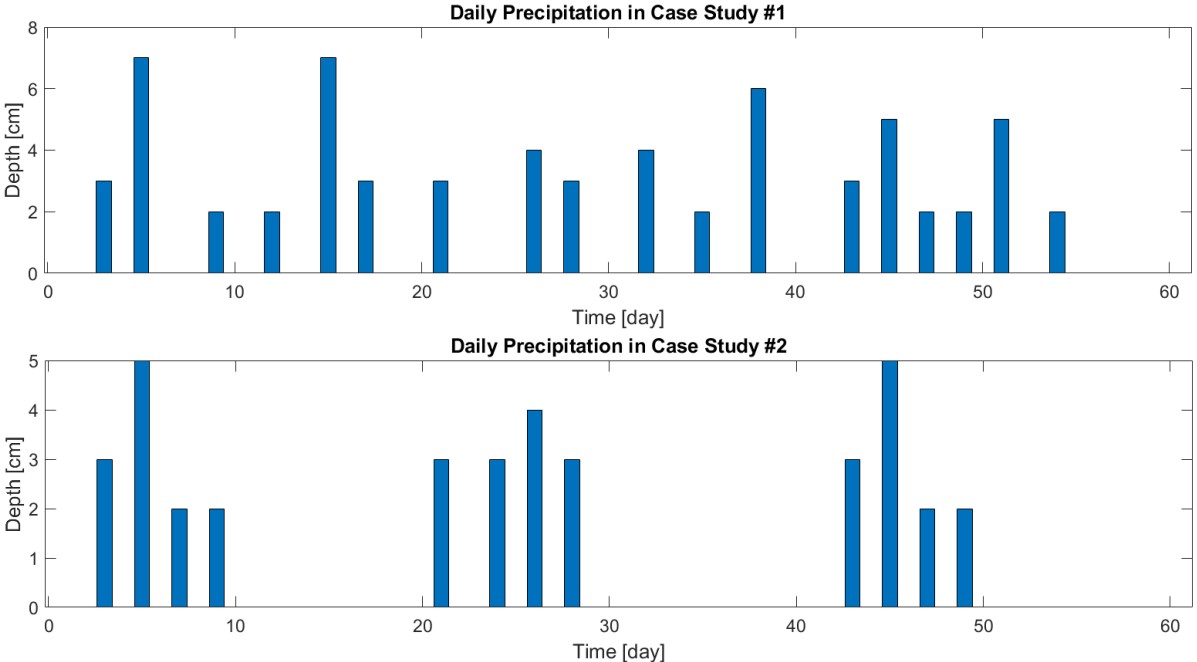

**Figure 2: Precipitation/irrigation (boundary condition) in Case Study #1 and Case Study #2.**

## 4 Results

### 4.1 Case study #1 – Random boundary condition

The ability of the model ensemble to predict accurately future variations of soil moisture depends directly on the accuracy of
the parameter estimates, which can be appreciated from Fig. 3. In this figure, the average of the parameters estimation
(posterior) at each time step is presented. In most of the time steps, the estimation of the parameters in C-GPFM is improved
compared to the biased (no assimilation) case. More importantly, the estimations of C-GPFM are superior and more
consistent than GPFM. This can be attributed to the fact that when a parameter is non-influential, C-GPFM does not involve
it in the assimilation process. For example, the parameters of the middle layer are less correlated with the measurements than
the other parameters, which was to be expected since measurements were performed only in the top and bottom layers.
Therefore, the estimation of the middle layer parameters by GPFM is erratic and inconsistent. By comparison, when
applying C-GPFM these parameters remain mostly constant as they are rarely involved in the data assimilation procedure.





Overall, the estimations of $\alpha$ and $\theta_{sat}$ were better than $K_{sat}$ in both methods due to the high influence of these parameters on the water content (Xu et al., 2016). However, the estimations of $\theta_{sat}$ by C-GPFM are superior to CPFM at the middle layer.

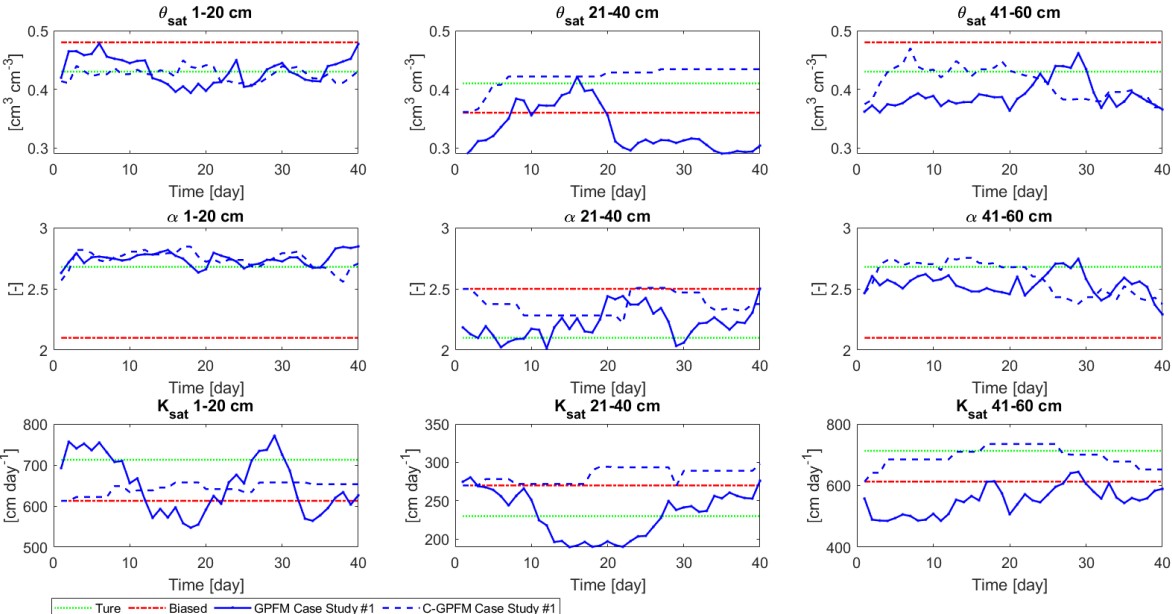

**Figure 3: Average of parameters posteriors by C-GPFM and GPFM together with the true and biased values for Case Study #1.**

The accuracy of the estimation of the parameters is reflected on the estimation of the current state, i.e., water content (Fig. 4). The top frame shows the absolute estimation error integrated over the whole soil profile (averaged over the 100 models of the ensemble) while the other frames show the absolute estimation error at the middle of each layer (averaged over 1 cm depth). The consistency and accuracy of the C-GPFM can be clearly observed in all three layers. The most noticeable improvement is observed in the second layer, which agrees with the previous observation regarding the "stability" of the parameter estimations in that layer (Fig. 3).

The superiority of C-GPFM in estimating the whole moisture profile is further demonstrated in Fig. 5, which shows snapshots at selected days. The main advantage of C-GPFM is at the mid-layer although C-GPFM is also slightly better at the other layers. In addition, the estimates obtained by C-GPFM are consistently improved, while the results obtained by GPFM are not consistent.

The ability of the data-assimilated models to predict accurately soil moisture evolution in response to future precipitation events was further tested as follows: At each assimilation day, the average ensemble of posterior parameters was used to predict the soil water content for the next 20 days. The results are shown in Fig. 6. Compared to GPFM, C-GPFM improved the prediction accuracy in the middle layer significantly, and this improvement remained quite stable throughout the whole period. By comparison, starting from day 20 there is a significant degradation of the predictions of GPFM. This agrees with





the degradation in the current state estimation observed in Fig. 4 and Fig. 5. The estimation for the top and bottom layers were good in all methods due to the improved estimations of the current state (Fig. 4) and parameters (Fig. 3).

**Figure 4: Average of state posteriors by C-GPFM and GPFM in Case Study #1.**



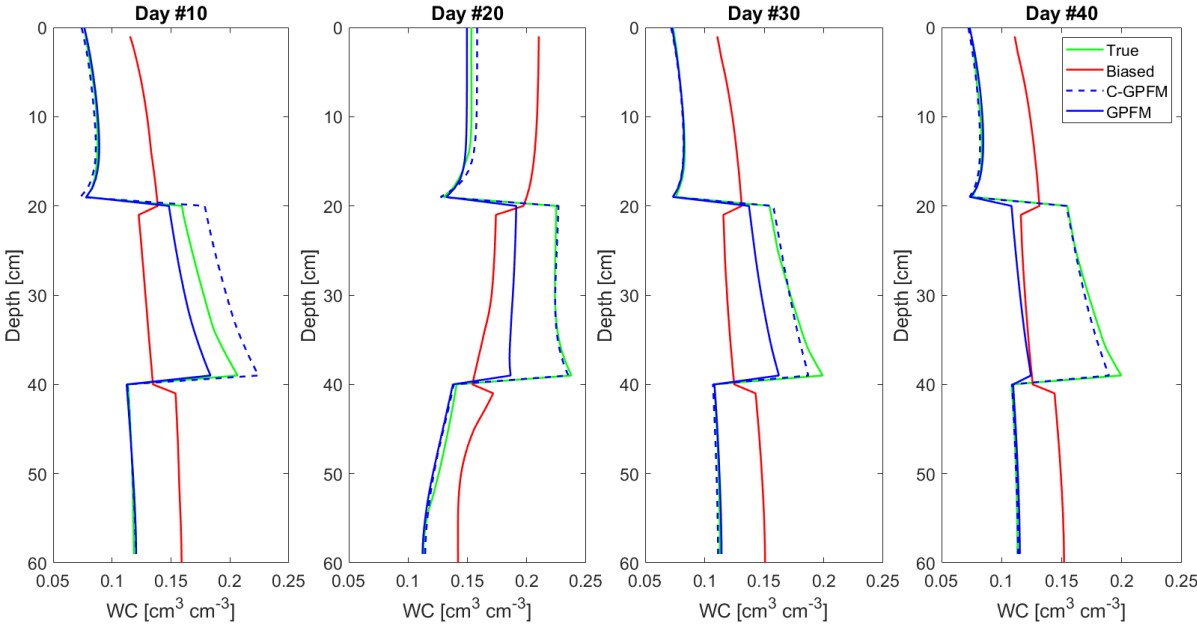

**Figure 5: Average of parameters posteriors by C-GPFM and GPFM at days #10, #20, #30 and #40 for Case Study #1.**

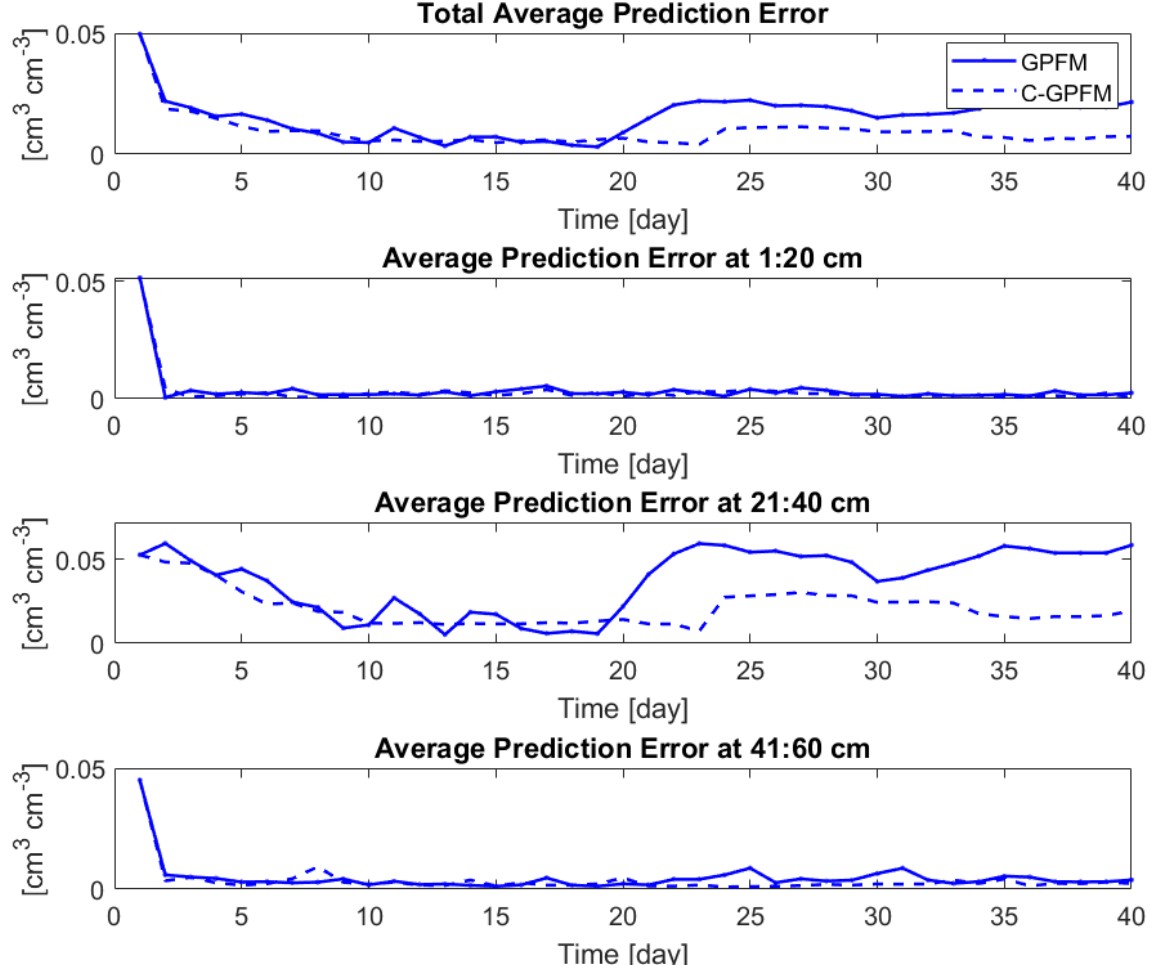

**Figure 6: Average of state prediction error for the next 20 days at each day of assimilation by C-GPFM and GPFM for Case Study #1.**

In order to neutralize the effect of the current state on future predictions and test solely the impact of parameters estimations on predictions, an additional test was conducted. In this test, at each assimilation day, the average of the set of the posterior parameters was used to predict the soil water content for a 20-day period. To test the robustness of the methods under different initial conditions and different irrigation schedules, one hundred runs were conducted, in which the initial conditions were set as the true values corrupted by white Gaussian noise with a standard deviation of 10%. In addition, the probability of precipitation on each day was set to 50% and the precipitation amount on those days was sampled from a uniform distribution between 0 and 10 cm. The error (averaged over time, depth and runs) of the ensemble at days 31, 33, 35, 37 and 39, normalized with respect to the biased model error on the same day is presented in Fig. 7. Compared to the biased model (no assimilation), both assimilation approaches (GPFM and C-GPFM) reduced the prediction errors by about 40%. However, C-GPFM is 5-10% more accurate than GPFM in all of the time steps. Fig. 8, which shows the percentage of





runs in which C-GPFM outperformed GPFM, shows that C-GPFM performed better than GPFM not only on average but also on the vast majority of single runs (about 80% of the runs).

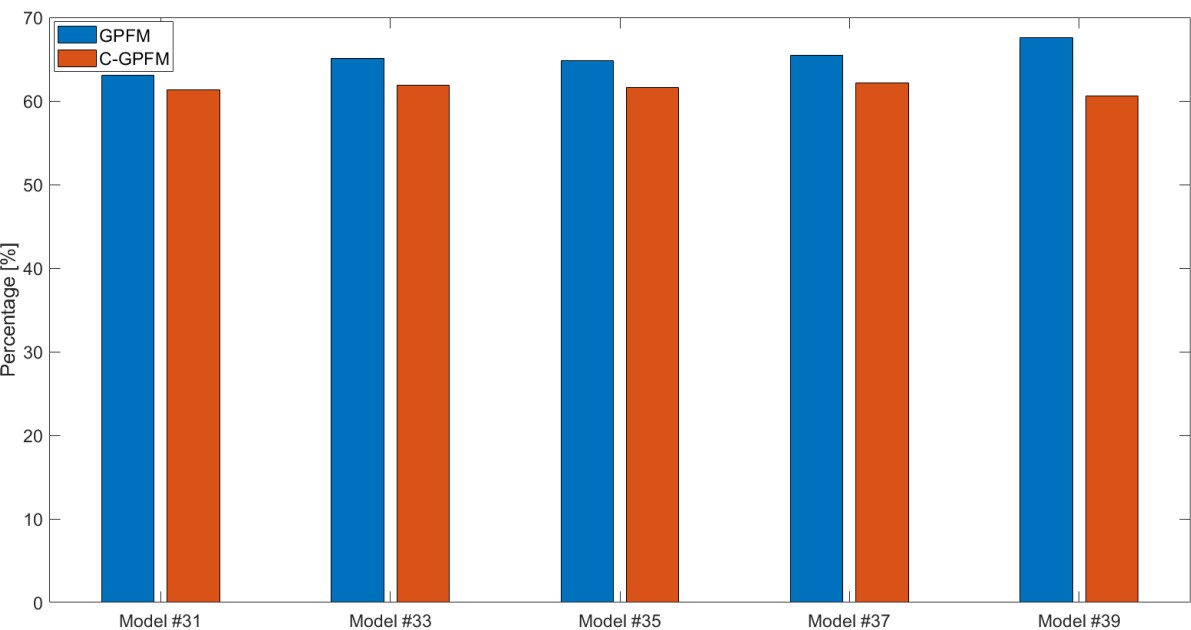

**Figure 7: State posterior errors (averaged over time, depth and runs) normalized with respect to the error of the biased (no assimilation) model. Results for the models obtained on days #31, #33, #35, #37 and #39 by C-GPFM and GPFM for Case Study #1.**

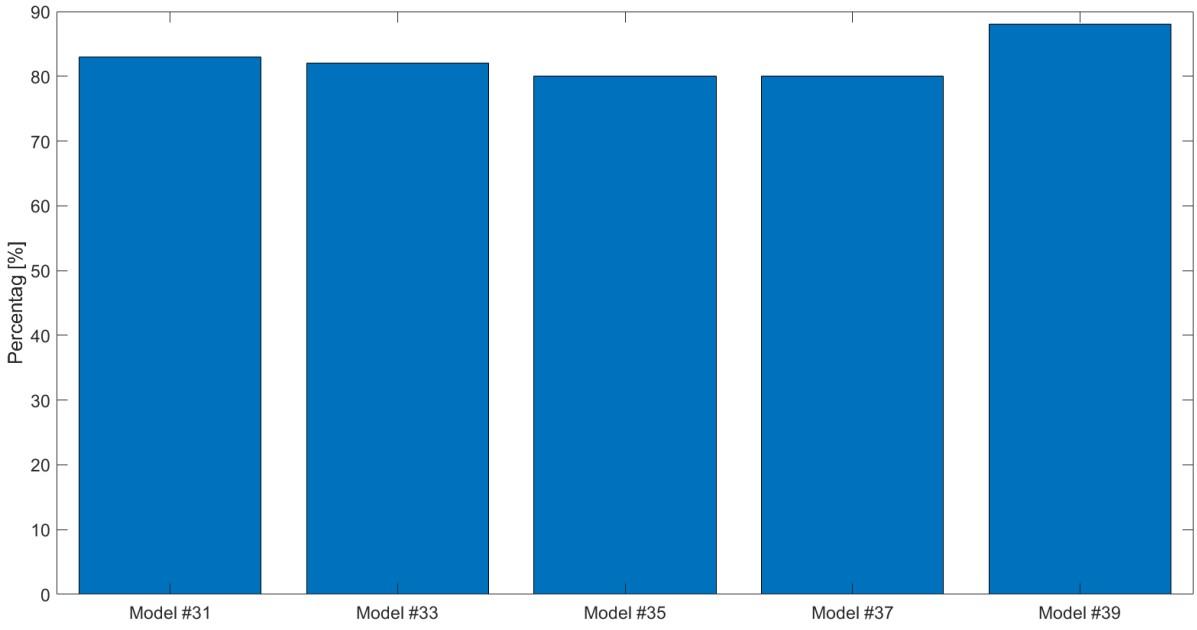



**Figure 8: Percentage of the runs in which C-GPFM led to smaller average state errors than GPFM. Results for the models**
**obtained on days #31, #33, #35, #37 and #39 for Case Study #1.**

The superiority of the C-GPFM over GPFM stems only from the dynamic selection of specific parameters according to their

correlation with the available measurements. In order to further investigate the relation between the accuracy of parameter

estimation and selection frequency (i.e. parameter influence), the relative absolute error of each parameter (averaged over

time) was plotted after sorting the parameters according to the number of times they were selected for adjustment (Fig. 9). In

addition, the number of times the parameter was selected for adjustment by C-GPFM is presented on the top of the figure.

The relative error obtained by GPFM is also shown for comparison. Overall, the parameters selected more frequently are

estimated more accurately than the parameters selected less frequently, which validates the motivation for the proposed

approach, namely that the stronger a parameter is correlated with measurements, the more the parameter is influential and the

more its estimation can be improved. As shown previously, the superiority of C-GPFM over GPFM in estimating most of the

parameters is clear.

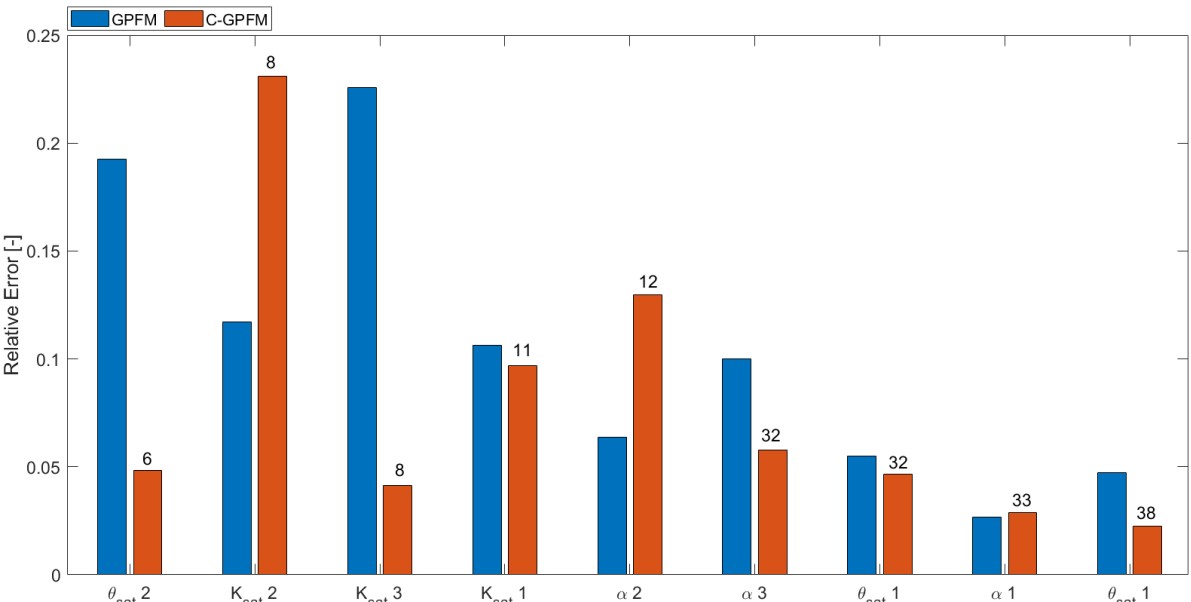

**Figure 9: Relative absolute error of each parameter (averaged over time) estimated by C-GPFM and GPFM together with the**
**number of times the parameter was selected for adjustment by C-GPFM for Case Study #1. The number of times a parameter was**
**selected for adjustment by C-GPFM is indicated above the corresponding sack.**

### 4.2 Case study #2 – cyclic boundary condition

The general trend of the results was similar to that observed in Case Study #1, and for brevity only the ability of the

ensemble to predict soil moisture in response to future wetting events is presented. More specifically, Fig. 10 shows the

average prediction accuracy of each ensemble when used to predict soil moisture for the next 20 days. Similar to Fig. 6, this





figure shows the overall prediction error as well as the prediction error for each layer, both for C-GPFM and GPFM. The superiority of C-GPFM, mainly on the mid-layer, agrees with the results of Case Study #1 (Fig. 6). In particular, it is interesting to observe the results of the models that have been adjusted using at least one full wetting-drying cycle, i.e. the models obtained from Day 20 onward: Clearly, the model obtained by C-GPFM is able to predict much more accurately the evolution of soil moisture during the second and third wetting-drying cycles.

**Figure 10: Average of state prediction error for the next 20 days at each day of assimilation by C-GPFM and GPFM for Case Study #2.**

## 5 Conclusion

This study presented a novel particle filter in which only a subset of the parameters was adjusted at each data assimilation

step. The selection of the parameters was achieved by correlation analysis rather than sensitivity analysis in order to avoid



high computation burden. The parameter-selection procedure was combined with the genetic - operators based PF with Monte Carlo Markov Chain (based on Jamal and Linker, (2020)), in such a way that only parameters highly correlated with the available measurements were involved in the selection, mutation, crossover and resampling operations of the PF. The uniqueness of the proposed method is the ability of identifying the highly influential parameters dynamically and in real-time

with only marginal additional computational costs. The proposed method was applied to a water flow model (Hydrus-1D) in which states (soil water contents) and parameters (soil hydraulic parameters) were updated via data assimilation. Overall, the proposed method yielded state estimates and parameter estimates that were more accurate and more robust (consistent) than those obtained when adjusting all the parameters at each data assimilation step.

**Author Contribution**

AJ wrote the code, prepared the case studies, analyzed the results, and wrote the manuscript. RL analyzed the results and reviewed the different versions of the manuscript.

**Competing interests**

The authors declare that they have no conflict of interest.

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
