# Peer review of "Covariance-based selection of parameters for particle filter data assimilation in soil hydrology"

_Hydrology and Earth System Sciences, 2021_

## Referee Comment (RC2)

Paper # hess-2021-295

**Covariance-based selection of parameters for particle filter data assimilation in soil hydrology**

**by**

*Alaa Jamal and Raphael Linker*

The paper presents a new particle filter data assimilation method where only highly correlated parameters to the measures are updated by the particle filter. A combination of Monte Carlo Markov Chain method and Particle Filter with Genetic Algorithm (Jamal and Linker, 2020) is used for the estimation of the state variables and parameters. The method is then applied to a water flow model where soil water contents and hydraulic soil parameters are updated using data assimilation. The results are compared to the traditional particle filter method where all parameters are updated during data assimilation.

**Comments:**

1/The investigated test cases are synthetic. The authors should consider simulation of real field or laboratory cases.

2/The constitutive law describing the relationships between pressure head, conductivity and water content should be specified when describing the test studies. Is it the Mualem–van Genuchten model?

3/The parameter $\alpha$ should have a dimension $[L^{-1}]$.

4/The value of the parameter n should be greater than 1

5/Why the parameter $n$ is fixed and not included in the inversion procedure? Both Mualem van Genuchten parameters $\alpha$ and $n$ cannot be measured and both should be included in the estimation.

6/The interval of variation +-40% is acceptable for the saturated water content, but not for alpha nor KS which can vary at least by one order of magnitude. Their intervals should be significantly enlarged because of the usually lack of prior knowledge of their values.

7/Why the estimation of the tetaS in the second layer is better with C-GPFM than with CPFM? Note that the inverse can be observed with KS ? How can you explain that parameters of this layer although they are not highly correlated to the measures, they can be better estimated with C-GPFM ? I believe that the parameters in the second zone cannot be accurately estimated with neither of the methods.

8/Could you provide more explanations and details why C-GPFM is more consistent than GPFM?

9/All presented results (for all figures) should include uncertainty ranges for the estimated parameters and variables and the results should be discussed based on these uncertainties.

In sum, the conclusion is not well supported by the analysis. I can understand that C-GPFM can be more efficient than CPFM but I really don't see why it could be more consistent or more accurate that CPFM. Further, the authors should consider real and not synthetic experiments.

I suggest major revisions.